# TEMPORAL RELEVANCE ANALYSIS FOR VIDEO ACTION MODELS

## ABSTRACT

In this paper, we provide a deep analysis of temporal modeling for action recognition, an important but underexplored problem in the literature. We first propose a new approach to quantify the temporal relationships between frames captured by CNN-based action models based on layer-wise relevance propagation. We then conduct comprehensive experiments and in-depth analysis to provide a better understanding of how temporal modeling is affected by various factors such as dataset, network architecture, and input frames. With this, we further study some important questions for action recognition that lead to interesting findings. Our analysis shows that there is no strong correlation between temporal relevance and model performance; and action models tend to capture local temporal information, but less long-range dependencies.

## 1 INTRODUCTION

State-of-the-art action recognition systems are mostly based on deep learning. Popular CNN-based approaches either model spatial and temporal information jointly by 3D convolutions (Carreira & Zisserman, 2017; Feichtenhofer, 2020; Tran et al., 2015) or separate spatial and temporal modeling by 2D convolutions (Fan et al., 2019; Lin et al., 2019) in a more efficient way.

One of the fundamental keys for action recognition is temporal modeling, which involves learning temporal relationships between frames. Despite the significant progress made on action recognition, our understanding of temporal modeling is still significantly lacking and some important questions remain unanswered. For example, how does an action model learn relationships between frames? Can we quantify the amount of temporal relationships learned by an model? Stronger backbones in general lead to better recognition accuracy (Chen et al., 2021; Zhu et al., 2020), but do they learn temporal information better? Do models capture long-range temporal information across frames?

In this paper, we provide a deep analysis of temporal modeling for action recognition. Previous works focus on performance benchmark (Chen et al., 2021; Zhu et al., 2020), spatio-temporal feature visualization (Feichtenhofer et al., 2020; Selvaraju et al., 2017) or saliency analysis (Bargal et al., 2018; Hiley et al., 2019b; Roy et al., 2019; Wang et al., 2018) to gain better understanding of action models. For example, comprehensive studies of CNN-based models have been conducted recently in (Chen et al., 2021; Zhu et al., 2020) to compare performance of different action models. Others (Monfort et al., 2019b; Selvaraju et al., 2017; Zhou et al., 2016) focus on visualizing the evidence used to make specific predictions, sometimes posed as understanding the relevance of each pixel on the recognition. In contrast, we aim to understand how temporal information is captured by action models, *i.e.,* temporal dependencies between frames or *how a frame relates to other frames in a video clip*.

In this work, we propose a new approach to evaluate the effectiveness of temporal modeling based on layer-wise relevance propagation (Gu et al., 2018; Montavon et al., 2019), a popular technique widely used for explaining deep learning models. Our approach studies temporal relationships between frames in an action model and quantify the amount of temporal dependencies captured by the model, which is referred to as *action temporal relevance (ATR)* here (Sec. 3.2). Fig. 1 illustrates our approach. We conduct comprehensive experiments on popular video benchmark datasets such as Kinetics400 (Kay et al., 2017) and Something-Something (Goyal et al., 2017) based on several representative CNN models including I3D (Carreira & Zisserman, 2017), TAM (Fan et al., 2019) and SlowFast (Feichtenhofer et al., 2018). Our experiments provide deep analysis of how

temporal relevance is affected by various factors including dataset, network architecture, network depth, and kernel size as well as the input frames (Sec. 4.2). Finally, based on the performed analysis, we effort to deliver a deep understanding of the important questions brought up above (Sec. 4.3). We exclusively focus on CNN-based approaches for action recognition. Nevertheless, we are fully aware that the recently emerging transformer-based approaches such as (Arnab et al., 2021; Bertasius et al., 2021; Fan et al., 2021b; Liu et al., 2021) demonstrate comparable or better performance than CNN-based models, but studying transformers is beyond the scope of this work. We summarize our contributions below:

- **Tool for Understanding Action Models.** We present a new approach for better understanding of action modeling and develop means of evaluating the effectiveness of temporal modeling.
- **Temporal Relevance Analysis.** We conduct comprehensive experiments to understand how temporal information in a video is modeled under different settings.
- **Deep Understanding of Temporal Modeling.** We study some fundamental questions in action recognition that leads to interesting findings: a) There is no strong correlation between temporal relevance and model performance. Instead, temporal relevance is more related to architectures. b) Action models behave similarly on both temporal and static actions defined by human (Sevilla-Lara et al., 2019), and there is no strong indication that temporal actions require stronger temporal dependencies learned by these models. c) As the number of input frames increases, action models capture more short-range temporal information (local contextual information), but less long-range dependencies. The better performance due to using more frames seems to be largely attributed to richer local contextual information, rather than global contextual information.

## 2    RELATED WORK

**Action Recognition Models.** Action recognition has achieved significant progress after the release of large-scale publicly available video datasets including Kinetics (Kay et al., 2017), Something-Something V2 (Goyal et al., 2017), Sports1M (Karpathy et al., 2014), Moments-In-Time (Monfort et al., 2019b), etc. Many models proposed different temporal modeling approaches to handle the temporal dynamics of a video (Carreira & Zisserman, 2017; Fan et al., 2019; Feichtenhofer, 2020; Feichtenhofer et al., 2018; Hussein et al., 2019; Lin et al., 2019; Luo & Yuille, 2019; Tran et al., 2015; 2019; Wang et al., 2020; 2016; 2018; Xie et al., 2018; Zhou et al., 2018). Chen et al. and Zhu et al. provided a comprehensive survey on how those CNN-based models achieve temporal modeling and compare their model accuracy. Transformer-based models have also become popular after their introduction to the computer vision community (Dosovitskiy et al., 2021). In addition, multiple recent attention-based temporal modeling works have been proposed to enhance the transformer-based models, *e.g.,* MViT (Fan et al., 2021a), TimeSformer (Bertasius et al., 2021), Video Swin (Liu et al., 2021), etc (Li et al., 2021; Neimark et al., 2021). The aforementioned works justify their capability of temporal modeling or the range of temporal modeling y validating the performance on benchmark datasets. In this work, we focus on quantifying the temporal relevance between each frame pair learned by a model to help understand the effects of different CNN-based approaches of temporal modeling for the task of action recognition.

**Model Analysis.** There are a few works that have assessed the temporal importance in a video, *e.g.,* Huang et al. proposed the method approaches to identify the crucial motion information in a video based on the C3D model and then used it to reduce sparse frames of the video without too much motion information; on the other hand, Sigurdsson et al. analyzed the action category by measuring the complexity at different levels, such as verb, object and motion complexity, and then composed those attributes to form the action class. Feichtenhofer et al. visualized the features learned from various models trained by optical flow to explain why the network fails in certain cases.

On the other hand, the receptive field is typically used to determine the range of a network can theoretically see in both spatial and temporal dimensions. Luo et al. thoroughly studied the spatial receptive field for image classification, and showed that the effective receptive field is much smaller than the theoretical one. In contrast, our work proposes an approach to quantify the learned temporal relevance between each frame pair, and uses this to understand how model architecture affects the temporal relevance.

**Explainability.**    Another popular research direction is to explain the decision made by a model through visualization of the class activation map, *e.g.,* CAM (Zhou et al., 2016), Grad-

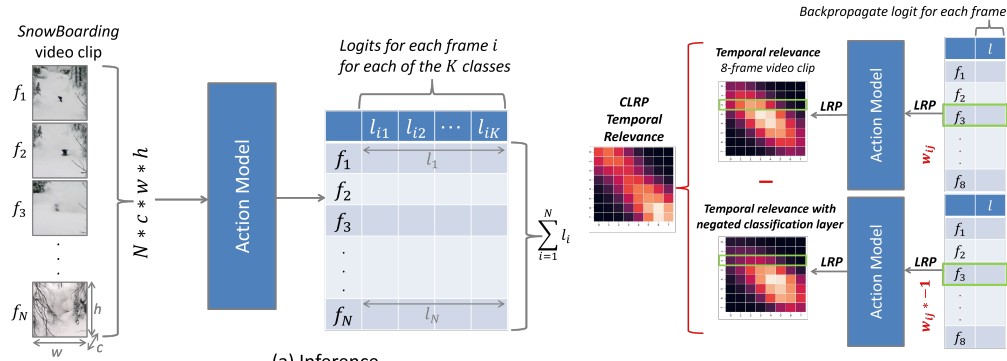

Figure 1: (a) A CNN takes a video with $N$ frames and outputs the average (or ensemble) predictions from all frames as the final prediction. (b) CLRP computes for an input frame the relevance scores of the target class and other classes by LRP, respectively. The positive differences of the two relevance scores represent how a frame is related to other frames. Temporal relevance is obtained to indicate the range of temporal dependencies of the frame on others.

CAM/GradCAM++ (Selvaraju et al., 2017; Chattopadhay et al., 2018), and layer-wise relevance propagation (LRP) (Binder et al., 2016) approaches. Binder et al. proposed the layer-wise relevance propagation (LRP) to show the relevance between each pixel and the predicted class; moreover, contrastive LRP further considers the effects of non-predicted classes to enhance the relevance to the targeted class (Gu et al., 2018). Several works extend LRP for spatio temporal explainable methods (Anders et al., 2019; Hiley et al., 2019a;b; Srinivasan et al., 2017). Hiley et al. surveyed a few frameworks that are used to explain action recognition models based on LRP or CAM approaches. Roy *et al.* (Nourani et al., 2020; Roy et al., 2019) add a probabilistic model on top of the action model to explain the meaning of each sub-action during the inference following Aakur et al.. Huang et al. study the effect of motion in action recognition by checking the accuracy drop from frames without using motion. Price & Damen explore frame-level contribution to the model output with Element Shapley Value. It should be noted that most of prior works try to find salient spatio-temporal regions for action recognition while our work quantifies how a model learns temporal relevances and investigate inter-frame relationship based on its architecture, as opposed to why a model makes a specific prediction.

## 3 METHODOLOGY

In this section we explain our approach to quantify temporal information captured by action models. We use the terms *temporal relevance* or the range of temporal dependencies for this. Our approach is based on layer-wise relevance propagation (LRP) (Binder et al., 2016) and contrastive layer-wise relevance propagation (CLRP) (Gu et al., 2018), which we briefly describe below. We develop a novel extension that enables CLRP to measure the temporal relevance between frames and apply it to analyze CNN-based action models.

### 3.1 CONTRASTIVE LAYER-WISE RELEVANCE PROPAGATION

LRP highlights which spatial or temporal input features are relevant to the predictions. LRP back-propagates signals from the prediction layer with pre-defined propagation rules. Relevance scores at layer $j$ ($R_j$) are propagated to the previous layer $i$ ($R_i$) in a neural network (employing a ReLU activation) following (Montavon et al., 2019):

$$R_i = \sum_j \frac{a_i w_{ij}}{\sum_{0,i} a_i w_{ij}} R_j, \tag{1}$$

where $a_i$ is the post-ReLU activation and $w_{ij}$ is the weights connecting layer $i$ to layer $j$. LRP back-propagates the logit $l_k$ of the target class $k$ to the input $X$ to get a relevance score $R = h_{LRP}(X, W, l_k)$. We use positive weights (*i.e.,* $z^+$ rule (Gu et al., 2018)) for convolutional and linear layers to remove noise in the gradients. Similarly, we apply the $z^\beta$ rule (Gu et al., 2018) to the input layer. We sum up all the pixel relevances in a frame to compute the frame-level relevance.

As discussed in (Gu et al., 2018), pixels with a high relevance value does not necessarily contribute to the target class only but to all the action classes. To mitigate the signals relevant to non-target classes, we extend contrastive layer-wise relevance propagation (CLRP) (Gu et al., 2018) for temporal analysis. Assume that we have weights $\boldsymbol{W} = \left\{ \boldsymbol{W}^1, \boldsymbol{W}^2, \cdots, \boldsymbol{W}^{L-1}, \boldsymbol{W}_k^L \right\}$, where $\boldsymbol{W}_k^L$ connects the $(L-1)$-th layer and the neuron for the class $k$. To offset signals for non-$k$ classes, we construct $\overline{\boldsymbol{W}} = \left\{ \boldsymbol{W}^1, \boldsymbol{W}^2, \cdots, \boldsymbol{W}^{L-1}, -1 * \boldsymbol{W}_k^L \right\}$ by negating the sign of $\boldsymbol{W}_k^L$, and compute $\overline{\boldsymbol{R}} = h_{LRP}(\boldsymbol{X}, \overline{\boldsymbol{W}}, l_k)$. Finally, CLRP is then defined as follows:

$$h_{CLRP} = \max \left( \boldsymbol{0}, \left( \boldsymbol{R} - \overline{\boldsymbol{R}} \right) \right), \tag{2}$$

## 3.2 ACTION TEMPORAL RELEVANCE (ATR)

It is generally believed that long-range temporal information is beneficial for action modeling. Accordingly, the temporal receptive field of a model is expected to cover all the relevant frames. For 3D convolutions, the activation of each neuron for a specific frame depends on the activations of neurons from other frames. We define the range of this temporal dependency on other frames as *action temporal relevance* (ATR). ATR is closely related to the effective receptive field (ERF) discussed in (Luo et al., 2016), but in the temporal domain. The spatial receptive field is largely dependent on the network depth and the temporal kernel size, but as pointed out in (Luo et al., 2016), the actual/effective receptive fields of a unit in 2D-CNNs, covers only a fraction of the theoretical receptive fields. In this work, we analyze the *effective temporal receptive field* for a video in temporal modules. Our approach also discovers the strength of temporal dependencies between frames. In addition, it should be noted that previous explainable methods on action modeling (*e.g.,* (Bargal et al., 2018; Ramanishka et al., 2017)) focus on finding spatially or temporally salient features while we aim to find the effective temporal range of a specific frame that is important for action classification.

**Temporal Relevance Computation.** For a CNN-based action model, it takes N sampled frames from a video as input and ensembles (or averages) the predictions of the $N$ frames as the final output (Fig. 1 (a)). We aim to obtain a temporal relevance matrix $A_{N \times N} = \{a_{ij} | i, j = 1 \cdots N\}$ computed from CLRP where $a_{ij}$ represents the relevance of the $j$-th frame to the $i$-th frame. Note that the temporal relevance score indicates how strong the temporal dependence is between two frames. Higher $a_{ii}$ (self-dependence) indicates shorter temporal dependence, where a model does not require other frames for its prediction,

Given an action model, for each input frame $f_i$, we compute the relevance score $a_{ij} = h_{CLRP}(f_j, \boldsymbol{W}, l_{ik})$ between the frame $i$ and $j$, where $\boldsymbol{W}$ is a set of model weights and $l_{ik}$ is the logit of the frame $f_i$ for the class $k$. Note that the summation of $i^{th}$ row of $A_{N \times N}$ is the logit of the given frame $f_i$ for the action class $k$, *i.e.,* $l_{ik} = \sum_j a_{ij}$, because LRP is conservative. We compute the ATR $r_i$ of a frame $f_i$ as the length of the shortest continuous segment $\{f_l, \cdots, f_i, \cdots, f_r\}$ with the sum of relevance scores that exceeds a certain amount of the total relevance score of the video, *i.e.,* $r_i = r - l + 1$ and $(\sum_{j=l}^{j=r} a_{ij})/(\sum_{j=1}^{j=N} a_{ij}) \geq \sigma$. We empirically set $\sigma$ to a high value, *i.e.,* 97.5% to cover relevant frames as many as possible.

We further define two metrics at video level: (1) avg-ATR, the average of the ATRs of all the frames weighted by their logits (only positive logits $l_i^+$ considered), *i.e.,* avg-ATR$= \sum_{i=1}^{i=N} w_i * r_i$ where $w_i = \frac{l_i^+}{\sum_{j=1}^{j=N} l_i^+}$ and (2) max-ATR, the maximum ATR among all the frames, *i.e.,* max-ATR$= max\{r_i\}$. Note the logit-based normalization is necessary as it removes the effects of negative and small logits and makes ATR comparable across videos (see Fig. 2). To summarize, the relevance matrix provides frame-level ATR (*i.e.,* effective temporal receptive field) while avg-ATR and max-ATR are video-level temporal relevance, which will be mostly used for our analysis. Moreover, we also use the avg-ATR/max-ATR for the dataset-level and class-level temporal relevance, which averages the video-level ATR accordingly. Higher ATR indicates higher temporal modeling capability of a model.

## 4 EXPERIMENTAL SETUP AND ANALYSIS

### 4.1 DATASETS AND MODELS

**Datasets.** We mainly use Kinetics-400 (*Kinetics*) (Kay et al., 2017), Something-Something V2 (*SSV2*) (Goyal et al., 2017) and MiT (Monfort et al., 2019a) in this analysis. *Kinetics*, arguably

the most popular benchmark dataset for action recognition, contains 400 activity classes; on the other hand, *SSV2* contains 174 classes for human-object interactions, and the videos are captured without strong background information. MiT contains 339 action classes including activity classes and human-object interactions. Most of our analysis in this work is based on models trained with the full *Kinetics* and *SSV2* datasets. Due to the high computational costs for model training, we also use mini-version of these datasets based on (Chen et al., 2021) for some experiments of this work, but only when necessary. In addition, we always compare models trained from full datasets or mini datasets for fairness.

**CNN-based Models.** In this work, we choose TAM2D (Fan et al., 2019) and I3D (Carreira & Zisserman, 2017) as a representative approach for 2D-CNNs and 3D-CNNs, respectively. While I3D is computationally inefficient, it still serves as the basis for many recent action recognition approaches such as SlowFast (Feichtenhofer et al., 2018) and TPN (Yang et al., 2020), and surprisingly performs on par with them, as shown in (Chen et al., 2021). Similar to (Chen et al., 2021), we remove all the temporal pooling layers in I3D as they hurt model performance. TAM2D is one of the efficient 2D-CNN models where temporal modeling is separated from spatial modeling. It is a generalized form of the popular TSM (Lin et al., 2019) and demonstrates strong performance on *SSV2*. Interestingly, the temporal aggregation module in TAM is indeed equivalent to 3D-depthwise convolution (Chen et al., 2021), which has been adopted by SOTA approaches such as X3D (Feichtenhofer, 2020) and CSN (Tran et al., 2019). We thus believe that studying these two representative video architectures can provide deep and helpful insight into how temporal dependencies are captured by action models. Additionally, we put an analysis of SlowFast in Appendix.

**Training and Evaluation.** We follow the training and evaluation protocols in (Chen et al., 2021) for both full datasets and mini-datasets. We adopt the *uniform sampling* to sample the frames from a video to model input. The uniform sampling first divides the whole video seqeunces into $F$ segments and then take one frame per segment to get a $F$-frame input. In evaluation, we use the single-clip setting for performance evaluation. More details can be found in Appendix.

### 4.2 TEMPORAL RELEVANCE ANALYSIS

All the models in our study based on *uniform sampling*. Compared to *dense sampling* that sees only a small portion of a video, uniform sampling covers significantly more of the video, so it is more suitable for studying long-range dependencies in temporal modeling. In Appendix, we provide a comparison of the two sampling strategies.

Our analysis contains a set of models trained with a different number of input frames (*i.e.,* 8, 16, and 32) and backbones (*i.e.,* ResNet18, ResNet50, ResNet101 and Inception-V1). A model is denoted by *X-Y-f[Z]* where $X$, $Y$, and $Z$ are the temporal module (*i.e.,* I3D or TAM2D), backbone network and input frame number, respectively. For example, I3D-R50-f8 indicates an I3D model using ResNet50 as a backbone and 8 frames as an input.

We compute the avg-ATR for video clips with correct predictions score $> 0.5$ and average the video-level avg-ATRs to obtain dataset-level ATRs. We first investigate how temporal modeling is affected by different datasets as well as various architecture-related factors including temporal module, backbone, temporal kernel size, and network depth, and the number of input frames. Based on this, we then focus on addressing several important questions of temporal modeling, which largely remain unclear in the field of video understanding.

**Dataset.** It is generally known that actions in *SSV2* indicate strong temporal dependencies while *Kinetics* actions are scene-dependent and less temporal. Our results in Table 1 confirm this nicely, consistently showing a larger average (and maximum) temporal relevance on *SSV2* than on *Kinetics* by I3D-R50.

Table 1: avg-ATR, max-ATR and model accuracy on different datasets. The red number indicates the different from *Kinetics*.

| Model | Acc. (%) | | avg-ATR | | max-ATR | |
|---|---|---|---|---|---|---|
| | *Kinetics* | *SSV2* | *Kinetics* | *SSV2* | *Kinetics* | *SSV2* |
| I3D-R50-f8 | 69.6 | 58.9 | 5.5±0.4 | 6.5±0.4 (+1.0) | 7.0±0.2 | 7.7±0.5 (+0.7) |
| TAM2D-R50-f8 | 70.5 | 60.2 | 4.2±0.4 | 6.0±0.4 (+1.8) | 5.2±0.5 | 7.0±0.3 (+1.8) |
| I3D-R50-f16 | 72.5 | 62.2 | 6.6±0.5 | 8.2±0.5 (+1.6) | 8.4±0.7 | 9.3±0.6 (+0.9) |
| TAM2D-R50-f16 | 73.1 | 63.0 | 4.9±0.4 | 7.3±0.6 (+2.4) | 6.1±0.8 | 8.3±0.7 (+2.2) |

Fig. 2 (a) and (f) further illustrate the temporal relevance heatmaps of the two datasets, which suggest that a frame is temporally related to only a few neighboring frames (*i.e.,* local dependencies). Nevertheless, the two datasets present striking differences in temporal modeling. Firstly, on *Kinetics*, all input frames make positive contributions to recognition, with more from the beginning and end frames of a video. Conversely, on *SSV2*, the frames in the middle play a more

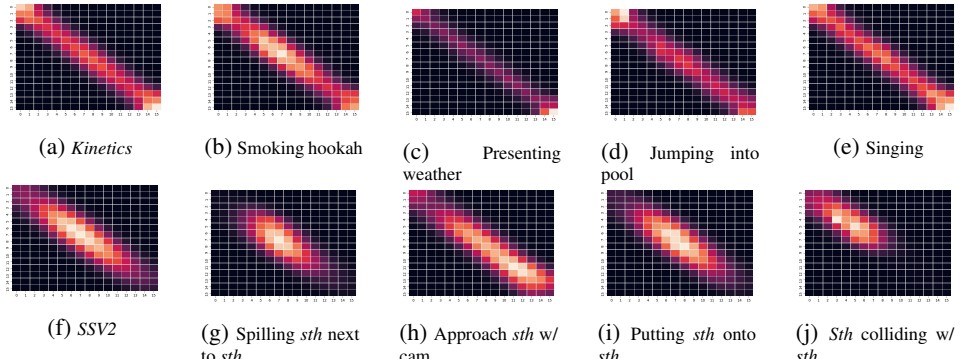

(a) *Kinetics*  (b) Smoking hookah  (c) Presenting weather  (d) Jumping into pool  (e) Singing

(f) *SSV2*  (g) Spilling *sth* next to *sth*  (h) Approach *sth* w/ cam  (i) Putting *sth* onto *sth*  (j) *Sth* colliding w/ *sth*

Figure 2: Heatmaps of dataset-level and class-level avg-ATR for I3D-R50-f16. A heatmap shows how each frame is relevant to other frames, with lighter colors denoting higher relevance. Column 1 demonstrates that models learn a larger avg-ATR on *SSV2* (f) than on *Kinetics* (a). The first and second rows depict avg-ATR of individual classes from *Kinetics* (b)-(e) and *SSV2* (g)-(j).

substantial role for recognition. Secondly, *SSV2* shows significantly more relevances on the diagonal, hinting that larger frame differences exist in the data than in *Kinetics*. Fig. 2 (b)-(e) and (g)-(j) show the heatmaps for a few different actions, which present noticeable variations between individual classes. Similar results are observed with TAM2D. On mini-MiT, we observe that its ATR is slightly higher than that of *Kinetics* but lower than that of *SSV2*. Due to space limitations, the results of mini-MiT can be found in Appendix.

**Temporal Module and Backbone.** The key difference between 2D-CNNs and 3D-CNNs resides in the fact that 3D-CNNs jointly learn spatial and temporal information. The temporal modules of action models are closely related to the backbones adopted by these models, and so we investigate the effects of both jointly. Fig. 3 shows how the temporal relevance differs between I3D and TAM2D using the same input frames and ResNet50 backbone. This clearly

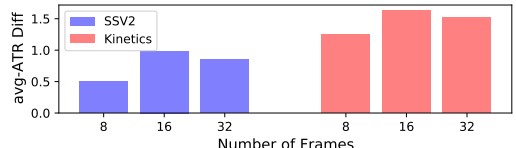

Figure 3: avg-ATR difference between I3D-R50 and TAM2D-R50, (avg-ATR(I3D) - avg-ATR(TAM2D)) at different number of frames. I3D presents larger temporal relevances.

indicates that I3D yields a larger relevance than TAM2D, and the gap is more significant on *Kinetics*. Since the temporal module of TAM2D is essentially a depth-wise 1D temporal convolution, it exchanges temporal information along the same spatial location at the same channel only. Differently, I3D uses a 3D convolution that allows for interactions in a larger scope. It is thus understandable that TAM2D is not as effective as I3D in capturing temporal dependencies.

On the other hand, stronger backbone networks in general lead to better spatio-temporal representations and action recognition performance (Chen et al., 2021). What is still unclear is whether a better backbone enables more temporal interactions for an action model.

In Fig. 4, we compare the avg-ATR captured by models trained on mini datasets using different backbones including Inception-V1, ResNet18 and ResNet50. The results are sorted by avg-ATR from high to low. As seen from the figure, while the backbone networks present a

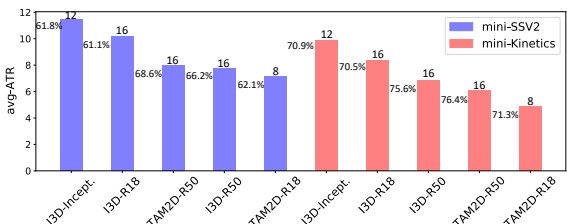

Figure 4: Effects of backbones on temporal relevance (16-frame models). The number on top of each bar indicates the number of temporal convolutions in a model while the number on the side is the model accuracy. This shows that avg-ATR does not correlate the strength of backbones.

similar order on two datasets, the order is somewhat random, and indicates no strong correlations with the strength of the backbones. By closely examining the temporal modeling in I3D and TAM2D, we find that their differences could result in a different number of temporal convolutions even with the same backbone, as shown by the numbers above the bars in Fig. 4. For example, I3D-R18 has 8 residual blocks, each of which has 2 3×3×3 convolutions, yielding a total of 16

temporal convolutions. On the other hand, TAM2D-R18 has one temporal module following each residual block, giving a total of 8 temporal convolutions only. The number of temporal convolutions seems to better explain the differences of temporal relevance between these models. Interestingly, I3D-Inception indicates the most effective temporal modeling capability with the largest temporal relevance. This is largely because of the $7 \times 7 \times 7$ kernel used at the beginning of the model, which greatly enhances I3D-Inception's ability to learn temporal information. As shown next, kernel size influences temporal modeling significantly.

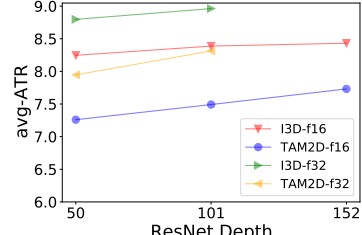

To summarize, there is no strong correlation between ATR and model performance. We also observe that the SlowFast shows a similar trend, but with a smaller avg-ATR than I3D and TAM2D due to architectural differences in Sec. B.

**Network Depth.** Fig. 5 shows that the avg-ATR is increased by network depth, but rather insignificant ($< 0.5$) considering the notable growth in network depth (more precisely, the number of temporal modules). We hypothesize that this has to do with the model ensemble effect discussed later in Sec. 4.3, which makes it less desired for a model to learn long-range temporal information.

Figure 5: Effects of network depth on *SSV2*. There is a very marginal increase of avg-ATR by network depth.

**Temporal Kernel Size.** The convolutional kernel size of a temporal module directly controls how many frames are involved in temporal modeling. Intuitively, it should have a significant direct impact on temporal relevance, which we have already seen in I3D-Inception. To further validate this, we experiment with temporal kernels of 3, 5, and 7 in TAM2D on mini-*SSV2* in Table 2. As expected, a larger kernel size produces larger temporal relevance, and the increase in the case of 16 input frames is quite pronounced, compared to the impact of network depth on temporal relevance. Nevertheless, this table shows again that there is no correlation between temporal relevance and model accuracy.

Table 2: Effects of temporal kernel size on mini-*SSV2*.

| Kernel size | TAM2D-R50-f8 | | |
|---|---|---|---|
| | Acc. (%) | avg-ATR | max-ATR |
| 3 | 65.4 | 6.3 | 7.4 |
| 5 | 64.1 | 7.8 | 8.0 |
| 7 | 63.6 | 8.0 | 8.0 |

**Input Frames.** CNN-based action models usually require a large number of input frames to achieve competitive performance (Carreira & Zisserman, 2017; Fan et al., 2019; Feichtenhofer et al., 2018; Chen et al., 2021). It would be an intuitive and reasonable thought that a strong model would be able to capture long-range temporal dependencies better in such a case. As demonstrated in Fig. 6, when increasing the number of input frames, the avg-ATR is indeed increased accordingly, but at quite a slow pace. Quadrupling the input from 8 frames to 32 frames only results in an increase of temporal relevance by ∼2 frames. Nevertheless, when looking at the temporal relevance over the video clip length, a model using more input frames learns shorter-range temporal information (also see Fig. 2), which is counter-intuitive. In what follows, we first validate our observations here, and then conduct further

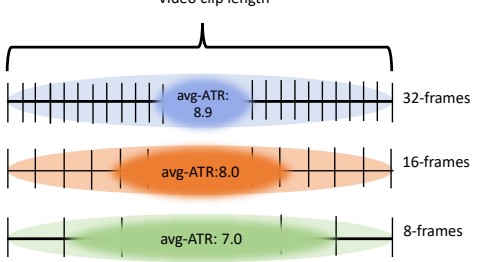

Figure 6: An illustration of the effect of input frames on temporal relevance of I3D-R50 trained on *Kinetics*. More input frames lead to larger temporal relevances (avg-ATR), *i.e.,* the absolute length of the highlighted frames. However, a model captures shorter-range temporal dependencies, *i.e.,* the relative length of the highlighted windows to the entire video clip length.

analysis of this in Sec. 4.3, wherein we relate this surprising finding to model ensembling. It should be noted that the long-range context we investigate here is within a single action and its temporal dependencies captured by models, not the long-term temporal information studied in (Wu et al., 2019) that is across actions in a video. We provide additional extensive and comprehensive analyses and results in the appendix.

Additionally, we evaluate the captured ATR from our approach with a partial uniform sampling approach in Sec. A of the Appendix. The experiment also indicates that only a portion of entire input frames captured by ATR suffice for an action prediction (Fig. 12 in the Appendix).

### 4.3 WHAT ARE ACTION MODELS LEARNING?

We have provided an in-depth analysis in terms of how temporal information is learned by CNN models and how architecture-related factors affect temporal modeling. Our approach is related to the effective receptive field (ERF) proposed in (Luo et al., 2016), in that the frame-level temporal relevance in our case can be considered as the ERF in the temporal domain. Our approach shows similar results to ERF with regards to kernel size, but our work focuses more on video understanding. In this section, we further examine some important questions related to temporal modeling for video understanding based on our analysis above.

**1) Do better-performing action models capture temporal dependencies better?** CNN-based approaches have made significant progress in action recognition and many of them claim to have achieved better spatiotemporal representations for video understanding. However, the question remaining unanswered is whether these approaches have actually enabled better temporal modeling. Based on the analysis above, our answer is probably No. For example, 1) TAM2D has lower avg-ATR than I3D in general, but its performance is on par with or better than I3D (Fig. 3); 2) I3D-Inception produces significantly larger

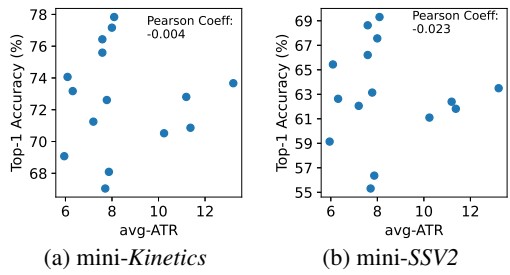

(a) mini-*Kinetics*          (b) mini-*SSV2*

Figure 7: avg-ATR vs. accuracy among different models based on mini datasets. The Peasron correlation coefficient indicates that there is no correlation between accuracy and avg-ATR.

temporal relevance than I3D-R50, but its performance is not as good as I3D-R50 (Fig. 5); and 3) larger kernels lead to larger relevance, but not necessarily accuracy increase (Table 2). We further plot model accuracy *vs.* avg-ATR for 15 models on mini-*Kinetics* and mini-*SSV2* in Fig. 7. The Pearson correlation coefficients are close to zero on both datasets, clearly indicating no correlation between accuracy and avg-ATR.

**2) Do action models learn temporal information differently by temporal dynamics of actions?** The temporal dynamic of an action suggests how much temporal information is needed to correctly recognize an action by a human. Temporal and static datasets annotated by humans from *Kinetics* and *SSV2* were constructed for temporal dynamic analysis (Sevilla-Lara et al., 2019). The *temporal dataset* consists of classes where temporal information matters while the *static dataset* includes classes where temporal information is redundant.

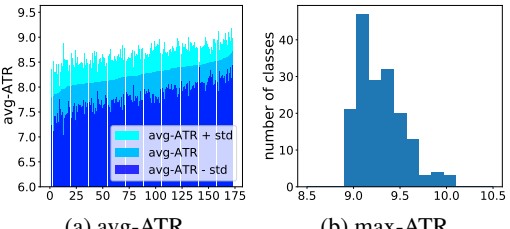

(a) avg-ATR          (b) max-ATR

Figure 8: (a) The distribution of avg-ATR and (b) the histogram of max-ATR on I3D-R50-f16 over classes for *SSV2*. Both ATRs do not show significant differences between classes.

It is then an interesting question to understand whether an action model requires (or relies on) more temporal information to classify temporal actions. In Fig. 8, we show the per-class ATR for *SSV2* classes (avg-ATR) in sorted order along with its standard deviation. For the majority of classes, their avg-ATRs are between 7.7 and 8.2, indicating no significant difference. Similarly, the difference of max-ATR is not greater than 1.0. This seems to suggest that an action model learns close amount of temporal information for most classes regardless of the temporal dynamics of actions. We investigate this question further by analyzing the temporal dynamics of action classes based on temporal relevance, *i.e.,* machine perception rather than human perception.

Specifically, we sort all the actions in a dataset from high to low by temporal relevance, and then pick the top-k and bottom-k classes as the most temporal classes and static classes, respectively.

Table 3 shows the overlap percentages of the temporal and static

Table 3: The class overlap of the temporal (T) and static (S) actions identified by human and machine.

| Model | avg-ATR | | | | max-ATR | | | |
|---|---|---|---|---|---|---|---|---|
| | SSV2 | | Kinetics | | SSV2 | | Kinetics | |
| | T | S | T | S | T | S | T | S |
| I3D-R50-f16 | 22.2% | 11.1% | 15.6% | 3.1% | 11.1% | 5.6% | 12.5% | 3.1% |
| TAM2D-R50-f16 | 38.9% | 16.7% | 18.8% | 6.2% | 27.8% | 11.1% | 21.9% | 6.2% |
| # classes | 18 | 18 | 32 | 32 | 18 | 18 | 32 | 32 |

datasets based on human and machine, respectively. Surprisingly, human and machine do not agree much with each other, especially on *Kinetics* and static classes.

We further plot the temporal relevance of the human-based temporal and static classes in descending sorted order in Fig. 9. It can be seen that both temporal and static classes are scattered, indicating the temporality of an action determined by a human is not a strong indicator of the temporal information needed by an action model for the action. In addition, there are many static classes (blue) with large temporal relevance.

Overall the large discrepancies from both datasets imply that the temporal

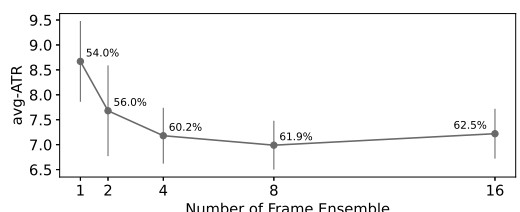

(a) *Kinetics*        (b) *SSV2*

Figure 9: Sorted avg-ATR of human-defined temporal and static actions (Sevilla-Lara et al., 2019) based on I3D-R50-f16. The scattered human-defined temporal (or static) actions suggest that the temporality of an action is not a good indicator of the temporal information captured by a model.

information perceived by a human as useful for the recognition might not be the same as what an action model attempts to learn. There is no strong indication that temporal actions require stronger temporal dependencies learned by a model (or static actions use less temporal information).

**3) What temporal dependencies are learned by action models?** Long-range temporal information in video data has long been considered crucial for action recognition. While many approaches (Wang et al., 2018) have attempted to capture such global dependencies across frames, but there is no easy way to understand whether the improved performance of these approaches is actually attributed to better temporal modeling. Our method analyzes the temporal dependencies.

Our analysis above points to a surprising finding that CNN-based approaches focus more on learning local temporal information rather than capturing long-range dependencies, indicated by a) the effect of input frames (Fig. 3); b) deeper models not resulting in (notably) larger temporal relevances (Fig. 5); and c) the evaluation based on partial uniform sampling (Fig. 12). We conjecture that this model behavior has to do with the frame ensembling previously shown in Fig. 1, which might be sufficient to provide long-range spatio-temporal features needed by an action model. We design a simple experiment to validate this. Specifically, we retrained a few 16-frame models by only taking

Figure 10: Effects of frame ensemble on *SSV2*. Models (TAM2D-R50-16f) are trained using different number of frames for ensembling during prediction. The top-1 accuracy of corresponding models are annotated. The avg-ATR increases as fewer input frames are used for prediction, suggesting that a model in such a case capture more temporal information for recognition.

1 frame or an average of evenly spaced 2, 4, and 8 frames as the model output. We expect as the number of frames used for ensembling is reduced, the model will resort to capturing more temporal interactions between frames to avoid dramatic performance degradation in such a case. As seen in Fig. 10, the temporal relevance increases as expected when fewer frames are involved in the ensemble. This experiment indicates that the better performance of a model using more input frames seems to be largely attributed to the ensemble of richer local contextual information learned by the model, rather than modeling global contextual information.

## 5 CONCLUSION

We have proposed an approach to quantify temporal modeling for action recognition. To the best of our knowledge, this is the first work that investigates the *effective temporal receptive field*, *i.e.,* action temporal relevance (ATR), in action models. Based on this, we conduct an in-depth analysis of how temporal information is captured for action recognition, which provides better understanding of action modeling. Contrary to the common belief that long-range temporal information can be captured by state-of-the-art action models, we observe that action models rely on short-range (local) temporal information, and capture less long-range (global) temporal information. We also find that a larger ATR does not necessarily lead to better accuracy.

**Code of Ethics and Ethics statement.** Our work is to analyze and investigate the temporal relevances of widely used temporal action models. We believe that there are no ethical concerns related to this work.

**Reproducibility.** We provide additional training details and LRP rules in Sec. F and G of the Appendix. Our implementation is based on the captum library (cap). We will make our code publicly available upon acceptance.

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

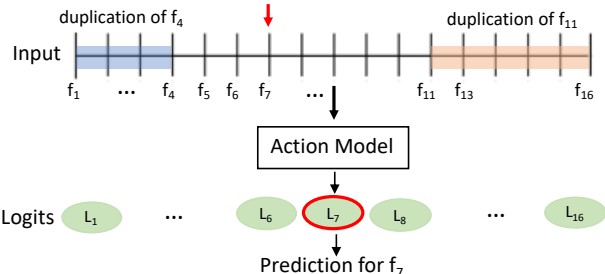

Figure 11: An illustration of partial uniform sampling. For a given frame $f$ (*e.g.*, $f_7$) in a window $W$ (*e.g.*, $f_4 - f_{11}$) from a set of uniformly sampled input frames, partial uniform sampling replaces the frames before $W$ by the start frame of $W$ (*i.e.*, $f_4$) and the frames after $W$ by the end frame of $W$ (*i.e.*, $f_{11}$). The modified input is fed into an action model to obtain the prediction for $f$.

**Appendix**

In Appendix, we provide more details and additional results which are not included in the main paper due to the space limit. We first explore partial uniform sampling for model evaluation in Sec. A. In Sec. B, we investigate a different backbone (*i.e.*, SlowFast) and dataset (*i.e.*, mini-MiT) with our proposed approach (Sec. B). We also compare CLRP with LRP. In Sec. C, we provide a comparison between dense and uniform sampling. We present more heatmap visualization in Sec. D and additional analysis on TAM2D in Sec. E. Finally, we give more details of model training protocols in Sec. F and show how to set up propagation rules in LRP in Sec. G.

## A    MODEL EVALUATION BY PARTIAL UNIFORM SAMPLING

Our analysis above suggests that action models does not capture long-range dependencies as expected when the number of input frames are sufficiently large ($\geq 16$). Fig. 2 strongly indicates that a frame is only temporally related to a few neighboring frames. This implies that these a few frames, rather than the entire input sequence, might suffice for the prediction from a frame in a model. In light of this, we propose a sliding-window method to validate the temporal relevance results presented above and help us understand temporal modeling better.

Recall that an action model usually generates a prediction for each frame separately and then ensembles the predictions of all the frames by averaging to obtain the final prediction (Fig. 1 (a)). We thus develop a *partial uniform sampling* strategy, which limits the prediction of a particular frame to only a portion of the entire input frames. The idea is illustrated in Fig. 11.

Specifically, for a target frame $f_i$ residing in a window $W$ of the input frames ranging between $[l, r]$ ($l \leq i \leq r$), we modify the input by replacing the frames before $l$ by $f_l$ and the frames after $r$ by $f_r$. This modified input is then fed into the model to obtain the prediction from $f_i$. Doing so only allows the prediction for $f_i$ to receive information from a *partial* portion of the input, *i.e.*, frames in the window $W$. We repeat this for each frame by sliding a window over the entire input and taking the average of all the predictions to be the final model output. Note that our method only keeps the prediction of one frame at a time and *the sliding window size can vary by frame*. The frame-level ATR in Fig. 12 shows the accuracy of this process. We average frame-level ATRs of all videos to represent the window size. This distinguishes it from the widely used multi-clip evaluation that takes an average of the predictions from all input frames in a clip.

We first experiment with a fixed sliding window size in partial uniform sampling to evaluate I3D and TAM2D models with 16 and 32 frames on both datasets. As shown in Fig. 12, the performance of all the models in the evaluation gets saturated quickly with around 8 frames on *Kinetics*, and 10 frames on *SSV2*, suggesting that frame-level prediction indeed does not use all the input. We further experiment with a dynamically changing window size based on ATR (*i.e.*, each frame based on a different window size defined by ATR), and the results (green and blue dots) match well with those based on a fixed window size. This confirms that measuring temporal relevance by our proposed approach is reasonable. We also tried empty images in partial uniform sampling, but the results are worse than using a frame image.

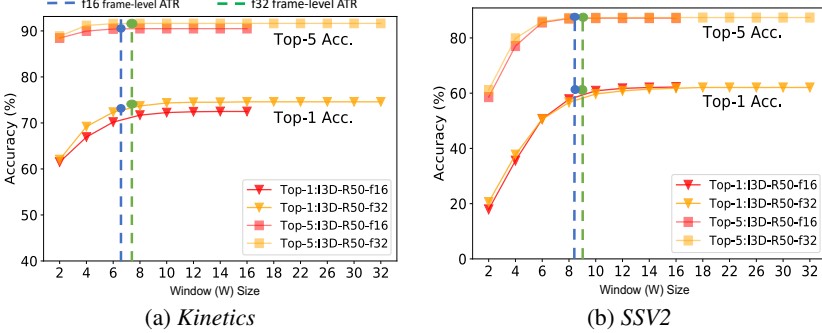

Figure 12: Model accuracy by window size using partial uniform sampling. The results show that frame-level ATR (blue and green dots) correctly indicates that only partial temporal information is needed for the prediction of a frame.

Table 4: avg-ATR, max-ATR and model accuracy on different datasets.

| Model | Single-clip Acc. (%) | | avg-ATR | | max-ATR | |
|---|---|---|---|---|---|---|
| | Kinetics | SSV2 | Kinetics | SSV2 | Kinetics | SSV2 |
| I3D-R50-f8 | 69.6 | 58.9 | 5.5±0.4 | 6.5±0.4 | 7.0±0.2 | 7.7±0.5 |
| TAM2D-R50-f8 | 70.5 | 60.2 | 4.2±0.4 | 6.0±0.4 | 5.2±0.5 | 7.0±0.3 |
| SlowFast-R50-8x8 | 68.9 | - | 3.7±0.7 | | 5.4±0.8 | - |
| I3D-R50-f16 | 72.5 | 62.2 | 6.6±0.5 | 8.2±0.5 | 8.4±0.7 | 9.3±0.6 |
| TAM2D-R50-f16 | 73.1 | 63.0 | 4.9±0.4 | 7.3±0.6 | 6.1±0.8 | 8.3±0.7 |
| SlowFast-R50-16x8 | - | 59.2 | - | 4.6±0.6 | - | 5.9±0.9 |

# B    ADDITIONAL RESULTS

**SlowFast Results.** The SlowFast video architecture (Feichtenhofer et al., 2018) is a dual-branch network with a slow branch processing frames at a low frame rate and a fast branch operating at a high frame rate. It is one of the best-performing models on Kinetics (Chen et al., 2021). To apply CLPR to SlowFast, we first merge the prediction of the slow branch into the fast one. Specifically, we expand the logits of the slow branch to the same size as the fast branch while keeping the logits at the slow frames unchanged and setting all others to 0. We then add the expanded logits to the logits of the fast branch, and perform CLRP on top of them for each frame, which produces a $n \times n$ relevance matrix $M$ where $n$ is the number of input frames to the fast branch. Additionally, it's observed that *the slow logits are dominant compared to the fast logits*, suggesting that the fast branch only plays a minor role in recognition. As a result, the relevance matrix $M$ is extremely uneven and sparse. To make our analysis meaningful, we sum up the relevance of every $r \times r$ block in $M$ (Sec. 3.2) to obtain an $m \times m$ matrix, where $m = n/r$ is the number of input frames to the slow branch. By doing so, we mainly focus on the slow branch in our analysis, but without neglecting the small but complementary contribution from the fast branch.

With the changes above in our method, we compute the ATRs for two SlowFast models based on *uniform sampling*, one model trained by ourselves on *Kinetics* using 8 slow frames and 32 fast frames (SlowFast-R50-8x8), and the other one trained by the authors of the original paper on *SSV2* using 16 slow frames and 64 fast frames (Facebook). Table 4 lists the results of these two models and their I3D counterparts, and Fig 13 further illustrates their relevance heatmaps. Interestingly, while SlowFast is fed with much more input frames, its ATRs on both *Kinetics* and *SSV2* are substantially smaller than those of I3D, suggesting that the fast branch do not strengthen model's temporal modeling ability as expected (Feichtenhofer et al., 2018). Closely examining the network architecture of SlowFast reveals that its special design aiming at computational efficiency may weaken its temporal modeling ability. Both branches of SlowFast eliminate the 7x7x7 covolution in I3D, and the slow branch introduces temporal convolutions only after the third layer, As illustrated in (Feichtenhofer et al., 2018). Due to that, the slow branch primarily focuses on spatial modeling, not temporal modeling. In addition, the lightweight fast branch seems too narrow to effectively capture the temporal dependencies in the input frames. Our analysis provides an explanation of why SlowFast underperforms on temporal datasets such as *SSV2* (Table 4).

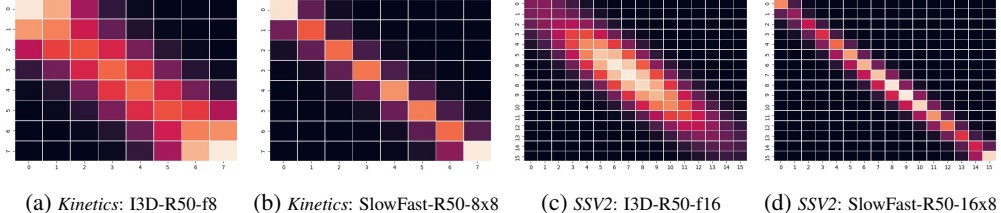

(a) *Kinetics*: I3D-R50-f8    (b) *Kinetics*: SlowFast-R50-8x8    (c) *SSV2*: I3D-R50-f16    (d) *SSV2*: SlowFast-R50-16x8

Figure 13: Heatmaps of dataset-level avg-ATR for I3D-R50 and SlowFast-R50. *Kinetics*: (a) and (b); *SSV2*: (c) and (d).

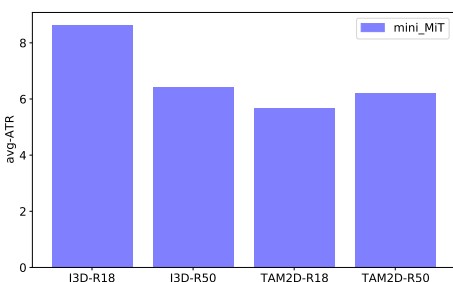

Figure 14: avg-ATR of different backbones trained with 16 frames on mini-MiT. We observe the similar trend in mini-MiT. I3D-R50 obtains lower avg-ATR compared to I3D-R18. At the same time, TAM2D-R50 obtains higher avg-ATR compared to TAM2D-R18.

**Results on mini-MiT.** We provide additional results on mini-MiT (Monfort et al., 2019b). In Fig. 14, we show avg-ATRs of different models using 16 frames. We observe a similar trend in mini-MiT to the Fig.4 in the main paper. I3D-R50 obtains lower avg-ATR than I3D-R18 while TAM2D-R50 produces higher avg-ATR than TAM2D-R18. Table 5 lists the avg-ATR and max-ATR of different models on mini-MiT, showing higher ATRs than those of *Kinetics* but lower than those of *SSV2*. This is probably because mini-MiT contains both human actions and human-object interactions, indicating intermediate temporality between *Kinetics* and *SSV2*.

**LRP Results.** Table 6 compares CLRP with LRP in terms of avg-ATR and max-ATR. In general, LRP obtains smaller avg-ATR and max-ATR than CLRP, indicating that by removing the signals related to non-target classes, the effective temporal range of input frames becomes longer in order to reproduce the original prediction.

## C    DENSE SAMPLING VS. UNIFORM SAMPLING

In comparison with *uniform* sampling where evenly-spaced frames across an video are extracted as the model input, *dense sampling* takes a consecutive set of frames of a given length from a video as the input. It is another sampling strategy widely used for action recognition, especially on the *Kinetics* benchmark. The densely sampled input frames are highly redundant and only cover a small portion of the video, thus in practice, dense sampling relies on the ensemble of the prediction results from multiple clips (10∼30) to achieve good results (multi-clip evaluation). In general, studying dense sampling for temporal analysis is less interesting. Here we provide a comparison of these two sampling strategies based on the *Kinetics* dataset.

Table 7 lists the avg-ATR and max-ATR of several I3D and TAM2D models based on uniform and dense sampling, respectively. We use the center clip of a video in dense sampling to compute ATRs. Note that the model accuracies shown here are generated by the single-clip evaluation not the multi-clip evaluation generally used for dense sampling. As seen from the table, dense sampling tends to produce smaller averaged temporal relevance (avg-ATR) than uniform sampling, but larger max temporal relevance (max-ATR). We further show the temporal relevance heatmaps of the models using 16 frames in Fig. 15. With uniform sampling, both I3D and TAM3D models demonstrate significant more contributions from the frames at the beginning and end of a video. This is reasonable as in a sequence of consecutive frames, the start and end frames are more likely to present a larger difference. In contrast, the models trained with uniform sampling indicate more contributions from the middle frames (Fig. 15-(a) and (b)).

Table 5: avg-ATR and max-ATR on mini-MiT.

| Model | avg-ATR MiT | max-ATR MiT |
|---|---|---|
| I3D-R50-f8 | 5.7±0.5 | 7.1±0.6 |
| TAM2D-R50-f8 | 5.5±0.5 | 7.0±0.4 |
| I3D-R50-f16 | 6.4±0.7 | 8.3±0.8 |
| TAM2D-R50-f16 | 6.2±0.6 | 7.0±0.8 |

Table 6: Comparison of action temporal relevance (ATR) between CLRP and LRP.

| Model | CLRP avg-ATR | | LRP avg-ATR | | CLRP max-ATR | | LRP max-ATR | |
|---|---|---|---|---|---|---|---|---|
| | Kinetics | SSV2 | Kinetics | SSV2 | Kinetics | SSV2 | Kinetics | SSV2 |
| I3D-R50-f16 | 6.6 | 8.2 | 6.3 (-0.3) | 7.8 (-0.4) | 8.4 | 9.3 | 7.3 (-1.3) | 8.9 (-1.0) |
| TAM2D-R50-f16 | 4.9 | 7.3 | 4.6 (-0.3) | 6.6 (-0.7) | 6.1 | 8.3 | 5.1 (-1.0) | 7.0 (-1.3) |

## D  ADDITIONAL VISUALIZATION OF HEATMAP

Fig. 16 demonstrates the heatmaps of more videos from *SSV2* and *Kinetics*. We also visualize the frames and their prediction logit scores shown by red color under each frame (the brighter the color, the higher the logit). Intuitively, predicting an action class in *SSV2* (Fig. 16-(a))) requires more frames as the action cannot be identified by a very few frames. Conversely, in *Kinetics*, most actions can be easily identified even with a few frames, as indicated by Fig. 16-(b). Especially, the examples of "doling lunadry" and "dancing ballet" do not present significant changes or movements in objects.

## E  ADDITIONAL RESULTS FOR TAM2D

**Model evaluation by partial uniform sampling on TAM2D.** Fig. 17 shows the model evaluation with partial uniform sampling. The finding here is similar to that in the main paper (Sec. A), i.e the performance of the TAM models gets saturated quickly around 8 frames on Kinetics and 10 frames on SSV2.

**ATR distributions over classes on TAM2D.** Similar to Fig. 8 in the main paper, Fig. 18 shows the per-class ATR for TAM2D-R50-f16. We observe a similar behavior to that in Fig. 8 of the main paper that the majority of classes do not have significant differences in avg-ATR in (a). In (b), most of the max-ATRs for classes are ranged from 9.0 to 10.0.

## F  TRAINING AND EVALUATION

Here we elaborate more how we train and evaluate models for our analysis. We follow the training and evaluation protocols in (Chen et al., 2021) for both full and mini-dataset. We adopt the *uniform sampling* to sample the frames from a video as the model input. The uniform sampling first divides the whole video sequences into $F$ segments and then take one frame per segment to get a $F$-frame input (random frame is used for training whole the center frame for inference). For the full datasets, the shorter side of a video is randomly resized to the range of [256, 320] but keeps the aspect ratio, then a random $224 \times 224$ spatial region are cropped along the time dimension. We trained the models with a batch size of 1024 by 128 GPUs for *Kinetics* and a batch of 128 by 16 GPUs for *SSV2*. More details can be found in (Chen et al., 2021). On the other hand, for the mini-dataset, we used the codes provided by (Chen et al., 2021) to train the models. During the evaluation, we use the single-clip setting for performance evaluation.

## G  DETAILS ON PROPAGATION RULES

Our implementation is based on the captum code (cap). We use the $z^+$ rule (Gu et al., 2018) for convolutional layers and fully connected layers in a CNN model, and the $\epsilon-$ rule (Montavon et al.,

Table 7: avg-ATR, max-ATR and model accuracy on *Kinetics*. The red number indicates the different from uniform sampling.

| Model | Acc. (%) | | avg-ATR | | max-ATR | |
|---|---|---|---|---|---|---|
| | uniform | dense | uniform | dense | uniform | dense |
| I3D-R50-f8 | 69.6 | 56.3 | 5.5±0.4 | 4.9±0.4 (-0.6) | 7.0±0.2 | 7.3±0.5 (+0.3) |
| TAM2D-R50-f8 | 70.5 | 56.3 | 4.2±0.4 | 4.3±0.3 (+0.1) | 5.2±0.5 | 6.2±0.8 (+1.0) |
| I3D-R50-f16 | 72.5 | 60.0 | 6.6±0.5 | 6.2±0.5 (-0.4) | 8.4±0.7 | 8.7±0.8 (+0.3) |
| TAM2D-R50-f16 | 73.1 | 59.2 | 4.9±0.4 | 4.8±0.5 (-0.1) | 6.1±0.8 | 6.5±0.9 (+0.4) |

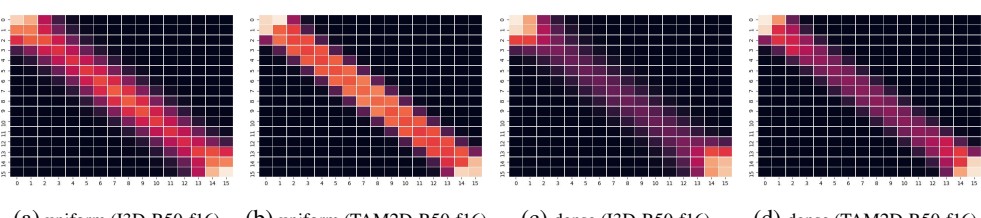

(a) uniform (I3D-R50-f16)  (b) uniform (TAM2D-R50-f16)  (c) dense (I3D-R50-f16)  (d) dense (TAM2D-R50-f16)

Figure 15: Heatmaps of mean avg-ATR for *Kinetics* models based on *uniform and dense sampling*. A heatmap show how each frame is relevant to other frames, with lighter colors denoting higher relevance.

2019) ($\epsilon = 1e - 9$) for max and average pooling layers. For batch normalization (BN), we apply the identity rule.

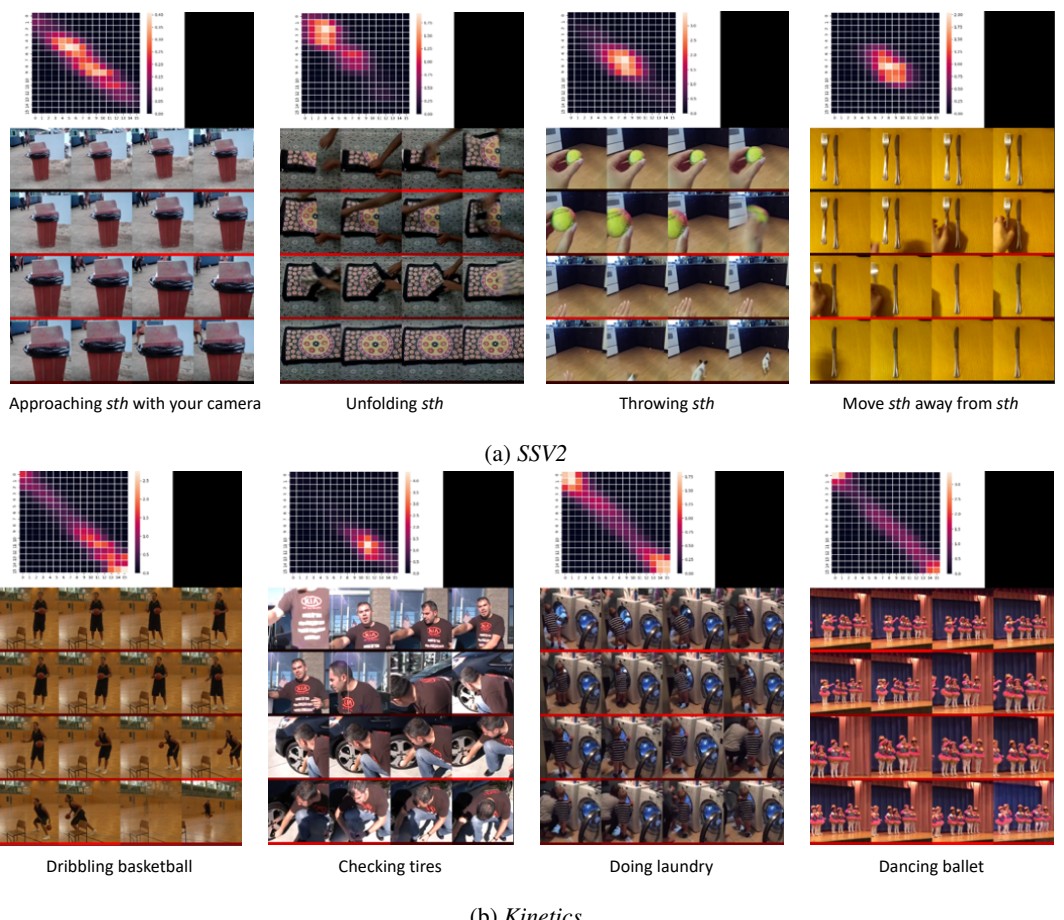

(a) *SSV2*

(b) *Kinetics*

Figure 16: Temporal relevance Heatmaps of actions in *Kinetics* and *SSV2*. The color bar under frame images represents the logit score of each frame for the predicted class. The brighter the color, the higher the score.

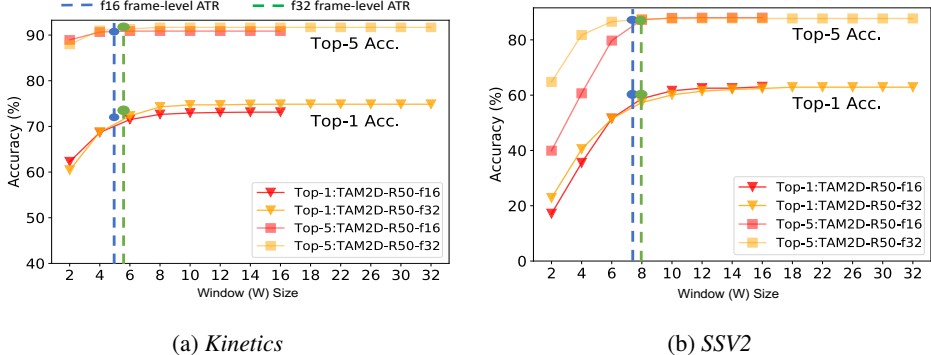

(a) *Kinetics*                                                    (b) *SSV2*

Figure 17: Model accuracy by window size using partial uniform sampling on TAM2D. The results based on frame-level ATR (blue and green dots) show that only partial temporal information is needed for the prediction of a frame.

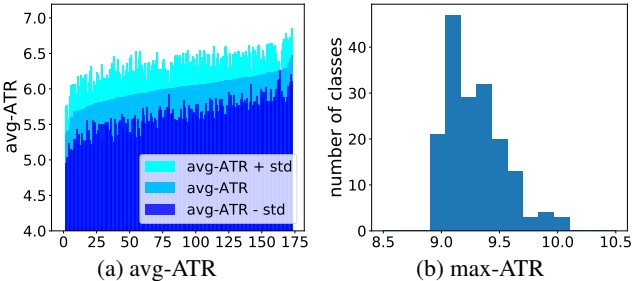

(a) avg-ATR  (b) max-ATR

Figure 18: (a) The distribution of avg-ATR and (b) the histogram of max-ATR over action classes on *SSV2* produced by the model TAM2D-R50-f16. Both ATRs do not show significant differences between classes.

