# OpenReview forum: "Temporal Relevance Analysis for Video Action Models"
_ICLR.cc/2023/Conference — Submitted to ICLR 2023_

### Official Review · Reviewer_4NU1 · 2022-10-24

**Confidence:** 4
**Correctness:** 2
**Technical Novelty And Significance:** 2
**Empirical Novelty And Significance:** 1
**Recommendation:** 3

**Clarity, Quality, Novelty And Reproducibility:**

+ Clarity: The techniques are not clearly explained and notions are confusing.

+ Novelty: This is in general an analysis paper. But some conclusions are not decisive (eg "no strong correlation", "no strong indication"). It is hard to tell what we can learn from the paper.

+ Reproducibility: the training and evaluation protocols follow publicly available resources. However, it is hard to reproduce due to the unclarity of the techniques.

**Details Of Ethics Concerns:**

no.

**Strength And Weaknesses:**

### Pros:

+  Understanding the temporal modeling for action recognition is an important task.

### Cons:

- The definitions of relevance scores are not clear.
  - How is the relevance score $\mathbf{\mathit{R}}$ accumulated from $l_k$? Also, should it make more sense to be $\mathbf{\mathit{R}}_k$ since it is class specific?
  - What is $z^+$ and $z^\beta$ rule? Any intuitive explanation?
  - The model to be analyzed employs a ReLU activation. What if other activations are used, eg. PReLU and GELU?
  - It is easy to get confused between $a_i$ in Eq (1) and $a_{ij}$ in the Temporal Relevance Matrix.

- ATR can be interpreted as a metric to measure the effective temporal receptive field. But the models that are studied in the paper have different *nominal* temporal receptive field (which can be directly obtained from the number of temporal convolutions being used along with their kernel size and stride). Therefore, does it make more sense to consider being normalized by the nominal temporal receptive field or consider effective and nominal temporal receptive field jointly?

- It is an somewhat surprising observation that video models with more input frames learn shorter-range temporal information. But how come the model does this? I don't see any further explanation..

- Many conclusions being drawn are either "no strong correlation" or "no strong indication". What can we learn from these conclusions? Or from an opposite perspective, should we conclude that the technique that is being used in the paper is not a good indicator to perform such temporal analysis??

**Summary Of The Paper:**

The paper proposes an approach to measure the temporal relevance in video models and studies the relationship between the performance of video model and its temporal modeling capability.

**Summary Of The Review:**

An analysis without many concrete conclusions. It is hard to tell what we can learn from the paper and how the conclusions can benefit designing a better video model.

---

### Official Review · Reviewer_6sPt · 2022-10-25

**Confidence:** 4
**Correctness:** 3
**Technical Novelty And Significance:** 2
**Empirical Novelty And Significance:** 3
**Recommendation:** 5

**Clarity, Quality, Novelty And Reproducibility:**

The paper is poorly written. It needs more work to get the final version. There’s confusion in presentations sometimes about architecture and plots. The subsections of analysis (network depth impact of a number of frames, different datasets) are not new directions, but, since it has never been looked at from this point of view, some sub-sections have surprising conclusions.


**Strength And Weaknesses:**

**Strengths**

- The authors have answered some unexplored questions about the temporal understanding of video action models. The analysis shows that backbone depth doesn’t have a significant impact, and that accuracy and avg ATR are inversely related most of the time.

- Experiment with Input frames shows an increasing number of frames doesn’t directly translate to learning information from more frames.

**Weakness**

-  The extension of CLRP is not clear. In the original work [1], Section 4, the authors discussed the concept of two ways to model non-target class. CLRP2 equation looks exactly the same as mentioned in the proposed extension approach (Section 3.1). It appears that the original CLRP work is used as it is.

- Temporal relevance Computation - Section 3.2 - The equation looks like only LRP is used for the ATR metric. Is it LRP or the CLRP?

- Analysis:

  - Selection of 3D CNN architecture - There could be a comparison table of performance with runtime and trainable parameters between a few architectures to justify the selection of I3D. There are several popular architecture which could have been used such as R3D and R21D are very popular architectures.

  - Datasets: Difference in Batch size Training  - Why K400 and SSv2 are trained with different batch sizes 1028 vs 128? Won’t that impact the numbers shown in Table 1?

  - Temporal Module - In Fig. 4, do different architectures have a different number of frames as input while training? If so, isn’t it unfair as in Figure 3, the number of frames matters, and the comparison is across different backbones here.

  - Fig. 5 - Missing values for R-152 backbones for I3D-f32 and TAM2D-f32.

  - Table 2 needs to be more detailed. It does not convey the whole story. What happens on the Kinetics dataset or a single dataset (like SSv2 in this case) different architecture?

- Learning of Action Model - 3rd point - Can the authors clarify what do they mean by “retrained a few 16-frame models”? Don’t the models train already on the K400 and SSv2 datasets?

- From all the analysis, could there be a recommendation to get the best performance out of the CNN-based temporal action model for both 3D and 2D?

**References:**

[1] Gu, Jindong et al. “Understanding Individual Decisions of CNNs via Contrastive Backpropagation.” ACCV (2018).



**Summary Of The Paper:**

The work focuses on the temporal analysis of CNN-based action recognition models. The authors proposed a new computation metric to compare the effective relevance field along the temporal dimension. The paper delves into different aspects such as dataset effects and variations in network backbone architecture, and finally answers some of the questions pertaining to action models for video understanding.

**Summary Of The Review:**

The extension of the metric already exists. Some of the experiment settings don’t look convincing to make the conclusion on those aspects. The paper still needs more work for certain sub-sections to have a conclusion on that analysis.

---

### Official Review · Reviewer_GyKL · 2022-10-25

**Confidence:** 2
**Correctness:** 4
**Technical Novelty And Significance:** 3
**Empirical Novelty And Significance:** 3
**Recommendation:** 6

**Clarity, Quality, Novelty And Reproducibility:**

The paper is well-written. Also, the paper proposed approach seems to be novel and useful. Considering the detailed description of the method, the quality and reproducibility of this paper are good.

**Strength And Weaknesses:**

Strengths
1. This is the first work that investigates the effective temporal receptive field, i.e., action temporal relevance (ATR), in action models.
2. This work well extends layer-wise relevance propagation (LRP) and contrastive layer-wise relevance propagation (CLRP) for quantifying the temporal relationships between frames for action recognition.
3. This work implements elaborate experiments to present the effects of various factors on temporal modeling and answer some important questions related to temporal modeling for action recognition. Some findings seem surprising compared to our intuition.

Weaknesses
1. The method proposed in this paper is based on existing methods, LRP and CLRP. The innovativeness in terms of methods is relatively weak.
2. Although this paper claims to focus on CNN-based methods, it is better to analysis the Transformer-based or pre-trained visual models. I wonder if it is able to apply the proposed techniques on the Transformer-based or Pre-trained models ?



**Summary Of The Paper:**

This paper proposes a method to quantify the temporal relationships between frames to effectively analyze temporal relevance for video action models. And then, comprehensive experiments present the effects of various factors on temporal modeling. Finally, based on experimental analysis, this paper answers some important questions related to temporal modeling for action recognition.

**Summary Of The Review:**

This work well extends existing methods for quantifying the temporal relationships between frames for action recognition. Comprehensive experiments are conducted to support the effectiveness of the proposed method. However, the innovation and generalization of the method need further consideration.

---

### Decision · Program_Chairs · 2023-01-20

**Decision:**

Reject

**Justification For Why Not Higher Score:**

- weaknesses as listed in summary
- no author response

**Justification For Why Not Lower Score:**

N/A

**Metareview: Summary, Strengths And Weaknesses:**

This work analyses the effective temporal receptive field of CNN models for action recognition.

Strengths:
+ propose new metric for comparing effective relevance field
+ investigates dataset effects and network backbones

Weaknesses:
- clarity
- incomplete or inconclusive analysis

**Summary Of Ac-Reviewer Meeting:**

N/A